# Sacral Bioneuromodulation: The Role of Bone Marrow Aspirate in Spinal Cord Injuries

**DOI:** 10.3390/bioengineering11050461

**Published:** 2024-05-06

**Authors:** José Fábio Lana, Annu Navani, Madhan Jeyaraman, Napoliane Santos, Luyddy Pires, Gabriel Silva Santos, Izair Jefthé Rodrigues, Douglas Santos, Tomas Mosaner, Gabriel Azzini, Lucas Furtado da Fonseca, Alex Pontes de Macedo, Stephany Cares Huber, Daniel de Moraes Ferreira Jorge, Joseph Purita

**Affiliations:** 1Department of Orthopedics, Brazilian Institute of Regenerative Medicine (BIRM), Indaiatuba 13334-170, SP, Brazil; josefabiolana@gmail.com (J.F.L.); dranapolianesantos@gmail.com (N.S.); luyddypires@gmail.com (L.P.); neurovirtual@gmail.com (I.J.R.); douglascoluna@hotmail.com (D.S.); tmosaner@uol.com.br (T.M.); drgabriel.azzini@gmail.com (G.A.); ffonsecalu@gmail.com (L.F.d.F.); alex_macedo@icloud.com (A.P.d.M.); stephany_huber@yahoo.com.br (S.C.H.); danfjorge@gmail.com (D.d.M.F.J.); 2Regenerative Medicine, Orthoregen International Course, Indaiatuba 13334-170, SP, Brazil; annu@navani.net (A.N.); jpurita@aol.com (J.P.); 3Clinical Research, Anna Vitória Lana Institute (IAVL), Indaiatuba 13334-170, SP, Brazil; 4Medical School, Max Planck University Center (UniMAX), Indaiatuba 13343-060, SP, Brazil; 5Comprehensive Spine & Sports Center, Campbell, CA 95008, USA; 6Department of Orthopaedics, ACS Medical College and Hospital, Chennai 600077, Tamil Nadu, India; 7Medical School, Federal University of São Paulo (UNIFESP), São Paulo 04024-002, SP, Brazil

**Keywords:** spinal cord injury, neuromodulation, orthobiologics, bone marrow aspirate, regenerative medicine

## Abstract

Spinal cord injury (SCI) represents a severe trauma to the nervous system, leading to significant neurological damage, chronic inflammation, and persistent neuropathic pain. Current treatments, including pharmacotherapy, immobilization, physical therapy, and surgical interventions, often fall short in fully addressing the underlying pathophysiology and resultant disabilities. Emerging research in the field of regenerative medicine has introduced innovative approaches such as autologous orthobiologic therapies, with bone marrow aspirate (BMA) being particularly notable for its regenerative and anti-inflammatory properties. This review focuses on the potential of BMA to modulate inflammatory pathways, enhance tissue regeneration, and restore neurological function disrupted by SCI. We hypothesize that BMA’s bioactive components may stimulate reparative processes at the cellular level, particularly when applied at strategic sites like the sacral hiatus to influence lumbar centers and higher neurological structures. By exploring the mechanisms through which BMA influences spinal repair, this review aims to establish a foundation for its application in clinical settings, potentially offering a transformative approach to SCI management that extends beyond symptomatic relief to promoting functional recovery.

## 1. Introduction

Spinal cord trauma is a complex injury that causes a series of disabling problems and functional deficits in patients [1]. Neurological injury is the most serious and debilitating alteration experienced by these patients. SCI interrupts nerve impulse conduction, affecting the ascending, descending and propriospinal pathways. This impairment can cause sensory, motor, proprioceptive or mixed deficits [2]. Such deficits can have devastating effects on the patient’s life, causing severe dependence on performing daily activities and personal hygiene [2]. According to the American Spinal Injury Association (ASIA), the loss of motor, sensory, or autonomic functions characterizes SCI, which can be complete or incomplete. Depending on the level of the injury, paraplegia may occur if it is below the level of the first thoracic vertebra (T1), or tetraplegia may occur if it is above this level [3].

The somatosensory pathway conducts our sensorial impulses to the thalamus and then to the somesthetic area, which is responsible for the body’s general sensitivity [4]. This area is located in the postcentral gyrus, corresponding to Brodmann areas, which may also be referred to as the primary somesthetic area or SI somesthetic area [5]. The hypothalamus and cerebral cortical regions process painful stimuli and respond to the release of inhibitory chemical and hormonal mediators, such as opioids, peptides, noradrenaline, and GABA, which can effectively suppress pain [6]. Coordinated locomotion, on the other hand, depends not only on the superior control of the cerebral motor cortex but also on the “neuromuscular memory” of the spinal cord by generating constant patterns and information to the interneurons to produce patterned behaviors [7]. Locomotion is a semiautomatic task designed for exploratory and escape behaviors and is driven by central pattern generators [8].

The development of neurological injury occurs in two stages: a primary phase caused by the trauma itself and a secondary phase caused by the inflammatory response to this trauma with changes in the extracellular environment [9]. These alterations culminate in an increase in the lesion, either due to ischemia associated with the inflammatory process or changes in pH, cytokine release, and ion flux [9]. Finally, an immune response will create a glial scar, which is an encapsulation of the injured area. Although this mechanism stabilizes injury, it also prevents neuronal growth, thereby impeding recovery from neurological deficits [9].

Additionally, the loss of central control as the communication between the brain and autonomic nerves becomes compromised leads to a condition known as autonomic dysautonomia or dysreflexia if the SCI occurs at or above the T6 level [10]. Consequently, there is a prevalence of sympathetic dominance, which means that the sympathetic nervous system operates excessively. Autonomic dysreflexia manifests as an exaggerated sympathetic response to stimuli below the injury level, potentially causing severe complications such as dysregulated blood pressure, increased heart rate, cerebral hemorrhage, myocardial infarction, and even death in more severe cases [11]. Cardiovascular reflexes become significantly impaired, contributing to difficulties in maintaining blood pressure within normal ranges. Conventional management strategies focus on identifying triggers, optimizing care, and sometimes using medications to address these dysregulations. Some of the medications used include antihypertensives, vasodilators, and alpha-adrenergic blockers to counteract excessive sympathetic activity and the associated cardiovascular issues [12]. Although these medications can be effective in managing cardiovascular problems, they may also have potential side effects. Adverse effects may vary, and their severity depends on factors such as the specific drug, dosage, and individual patient response.

The prolonged administration of certain medications may be associated with known side effects, including fatigue, dizziness, electrolyte imbalances, and, in rare cases, more serious side effects, such as kidney dysfunction [13]. Another significant concern is the recurring annual costs of caring for patients with chronic SCIs, which places a major economic burden on health care systems [14]. These costs include both direct medical expenses and indirect costs associated with the impact on the individual’s quality of life and productivity [15]. According to a previous study [14] conducted in the United States, the total annual direct medical costs for 675 patients with SCIs surpassed USD 14.47 million, corresponding to an average of USD 21,450 per patient. The average annual total costs for SCI patients vary, with complete cervical SCI incurring an average cost of USD 28,334 and incomplete thoracic SCI incurring an average cost of USD 16,792. Among the 675 patients included in the study, 233 were discharged at 378 hospitals, with a total cost exceeding USD 7.19 million. Additionally, outpatient care costs for the entire sample approximated USD 7.28 million. This underscores the importance of comprehensive health care for individuals with spinal cord injuries but also raises public health costs and concerns associated with long-term pharmacological drug administration.

The main motivation for the development of this study is based on the favorable results that researchers have obtained with the use of autologous bone marrow-derived products, such as BMA [16]. The bone marrow has an extremely rich molecular component with cytokines and chemokines that orchestrate the recruitment, modulation, and anabolic shift of the microenvironment, improving the regenerative process [17]. It contains a wide array of cell populations that can be divided into nonhematopoietic cells, including pericytes, endothelial cells, osteoblasts, adipocytes, and Schwann cells, and hematopoietic cells, such as neutrophils, lymphocytes, megakaryocytes, monocytes, and osteoclasts [18]. The bone marrow, however, is renowned for the presence of hematopoietic stem cells (HSCs) and mesenchymal stem cells (MSCs). These cells are highly praised due to their ability to self-renew and differentiate into specific mature cell lineages [19].

MSCs are known to manipulate their immediate microenvironment and modulate the wound-healing cascade, reducing apoptosis and fibrosis, dampening inflammation, and stimulating cell proliferation and differentiation via paracrine and autocrine pathways [20]. These cells release key molecules (Table 1), such as vascular endothelial growth factor (VEGF), transforming growth factor beta (TGF-β), stromal-derived factor 1 (SDF-1), and stem cell factor (SCF). Additionally, they can downregulate the synthesis of major proinflammatory cytokines, including interleukin 1 (IL-1), IL-6, interferon-γ (IFN-γ), and tumor necrosis factor-α (TNF-α) [21]. MSCs also exert immunoregulatory effects by impeding the activation of type 1 macrophages (MΦ1), natural killer (NK) cells, and B and T lymphocytes [22].

For these reasons, specifically, the application of this autologous orthobiologic material can improve the patient’s advantage and control the state of exacerbated inflammation in the extracellular environment of the injured area. It is possible to induce the sprouting of neurites in the presence of anabolic growth factors and prorepair type 2 macrophages.

Newly formed neurites may continue to develop into new axons or at least contribute to the repair of damaged tissue to reestablish the connections interrupted by trauma or make new connections with the sacral plexus (Figure 1).

This may create new neural networks that will overlap with the injured ones and reestablish links between the central and peripheral nervous systems. The abundance of bioactive molecules and the diverse cell populations found in BMA provide indispensable support to the extracellular environment after SCI. The formation and maintenance of neuronal networks connected to the sacral plexus may result in functional improvements in these patients. Currently, the literature remains very scarce in regard to the number of clinical trials specifically evaluating BMA for spinal cord injuries, which raises the level of relevance of this investigation.

Various studies demonstrated the safety and potential efficacy of autologous Schwann cell and mesenchymal stromal cell co-transplantation in improving neurological outcomes in chronic and complete SCI patients [30,31,32,33]. Current clinical trials continue to evaluate the effectiveness of autologous bone marrow stem cell transplantation in subacute SCI patients, aiming to provide further evidence on neurological function improvements and quality of life enhancements post treatment [34]. Moreover, comprehensive case studies have shown that the administration of autologous bone marrow stem cells via multiple routes is not only safe but also significantly improves the quality of life for SCI patients, underscoring the versatility and potential of stem cell therapies in real-world clinical settings [35]. The current review builds upon these foundational studies by focusing specifically on the biological neuromodulation of the sacral plexus, an area that has not been extensively covered in previous research. Unlike earlier works that have primarily concentrated on general recovery and the reduction of lesion size, our approach investigates the role of BMA in fostering the sprouting of neurites and the formation of new axonal connections specifically at the sacral plexus level. This focus is predicated on recent findings that suggest a pivotal role for the sacral plexus in lower limb functional recovery post SCI. Additionally, we propose a novel therapeutic framework that integrates BMA with other orthobiologic interventions, aiming to create a synergistic effect that could significantly enhance functional recovery. This nuanced approach not only addresses a critical gap in the literature but also proposes a targeted strategy that could potentially transform the management paradigms for patients with paraplegia or lower limb dysfunctions resulting from SCI. By detailing the specific contributions and the innovative focus of our review, we underline the relevance and urgency of this research in the broader context of SCI treatment advancements.

The objective of this manuscript is to discuss the mechanisms of biological neuromodulation of the sacral plexus based on the documented applications of bone marrow-derived products and propose them as feasible orthobiologic tools for the management of spinal cord injuries.

## 2. Pathophysiology of Spinal Cord Injuries

The severity of spinal cord injury depends on the level of injury. The greater the level of injury, the more challenging the rehabilitation process, with the need for training and multidisciplinary effort so that the patient can perform daily activities without assistance.

Spinal neurons involved in locomotion provide direct synaptic information to sympathetic preganglionic neurons and may suggest that locomotor function is intrinsically integrated with sympathetic autonomic functions at the T1-L2 spinal cord level [36]. For example, a spinal cord injury at T12 would result in the paralysis of the lower limbs but would keep the sympathetic system intact, which would retain the ability to activate sympathetic preganglionic neurons, whereas an injury above T1 would result in quadriplegia and the inability to activate the thoracic sympathetic nervous system [37]. The severity of injury is potentially determined by the number of supraspinal descending and ascending fibers, the neurons in the gray matter at the injury site that remain intact, and the level of injury within the spinal cord [38]. The ascending control of the lumbar spinal segments can coordinate and integrate movement without the brainstem spinal centers [39]. Few studies have examined the targets of projections from propriospinal neurons, much less the characterization of integration between spinal neurons or circuits responsible for locomotor and sympathetic integration in the upward direction.

At the molecular level, SCIs provoke an inflammatory response mediated by immune cells that migrate to the area of injury [40]. Subsequently, there is an increase in the number of various cytokines and inflammatory biomarkers, especially TNF-α (tumor necrosis factor alpha), which, in combination with its main receptors TNFR1 and TNFR2, appears to be involved in virtually all inflammatory responses [41]. This cytokine is also present in spinal cord injuries, as several studies have indicated [42,43,44,45,46,47,48,49]. This finding is highly relevant since TNF-α is one of the major causes of discomfort and neuropathic pain in patients with neurological injuries [50]. It is important to highlight the neurovascular connection between the spinal cord and the circulatory system [48]. The occurrence of SCI can cause significant alterations in the permeability of the barrier between the blood and the spinal cord, which may partially explain the extravasation and potential detection of associated inflammatory cytokines in the bloodstream [48].

TNF-α-induced neuropathic pain in humans is related to its role in regulating pain signaling through the glial system [51]. In the central nervous system, neurons, including astrocytes, oligodendrocytes, and microglia, are present in fewer numbers than glial cells are [51]. More specifically, spinal astrocytes contribute to chronic pain sensitization by activating the p38-MAPK signaling pathway [52]. Microglia are also involved in neuropathic pain, as they differentiate into macrophage-like cells in response to nerve damage and inflammation [53]. These cells, in turn, express antigens of the major histocompatibility complex and secrete proinflammatory cytokines such as TNF-α, IL-1, IL-6, CCL2, CX3CL1, and ATP, regulating their effects through the p38-MAPK signaling pathway [54,55,56]. From a biological perspective, in SCIs, TNF-α ultimately establishes a positive feedback loop in which it acts as a central regulator of inflammatory mechanisms in the glial system, further increasing its own production.

TNF-α and the TNFR1 receptor are responsible for catabolic processes, as they are linked to the degradation of the microenvironment, aggravating injury [57]. This cytokine may exert both protective and harmful effects, especially in SCIs [48]. TNFR1 is associated with neurodegeneration; neuronal TNFR1 specifically increases demyelination and exacerbates microglial inflammation [58], indicating that inflammation and chronic pain after spinal cord injury may be mediated via the elevated expression of this receptor. Furthermore, TNF-α and other cytokines also regulate the expression of proteolytic enzymes such as matrix metalloproteinases (MMPs) [59]. The exposure of nervous tissue to TNF-α alone induces vigorous MMP-9 expression by Schwann cells, an event that appears to mediate pain behavior [60]. This particular enzyme, also known as gelatinase B, degrades components of the basal lamina, contributing to the disintegration of the blood–spinal cord barrier (BSCB), leukocyte influx, oxidative stress, demyelination, and apoptosis [59].

Denervation is a typical consequence of SCI and often leads to muscle atrophy and decreased quality of life with a poor recovery prognosis [61]. Atrophy may involve the weight loss of individual fibers without reducing their total number or severe damage with necrosis in prolonged denervation [62]. In damaged muscles, fiber number and diameter may decrease within 9 months, while intramuscular adipose and connective tissue increase [63]. SCI patients with injuries below T10 or injury affecting the cauda equina region may suffer significant muscle denervation, leading to extensive muscle atrophy, intramuscular fat infiltration, and the formation of fibrous tissue. Such morphological alterations may increase the risk of developing other complications, including cardiovascular diseases, diabetes, obesity, and osteoporosis. Although muscle size may not change, a promising sign is the emergence of new fibers, especially within 6 months [63]. Unfortunately, long-term reinnervation after severe nerve injury often lacks regenerative effects substantial enough to compensate for muscle atrophy [64]. There are still no standardized neuromuscular rehabilitation protocols available to rescue muscles or restore their size in SCI patients, particularly those with motor neuron denervation of the lower limbs [65].

Emerging research suggests that BMA may offer a promising intervention in the complex pathophysiology of SCI, particularly through the modulation of the inflammatory and immune responses that characterize the secondary phase of injury [66,67,68]. BMA contains a rich array of MSCs and hematopoietic stem cells (HSCs), which have been shown to secrete a variety of bioactive molecules that can significantly modulate inflammation, promote angiogenesis, and facilitate the regeneration of damaged neural tissues. Specifically, MSCs have been identified to downregulate pro-inflammatory cytokines such as TNF-α, IL-1, and IL-6, which are pivotal in the inflammatory cascade following SCI. By reducing these cytokines, BMA may help attenuate the inflammatory response and mitigate secondary damage to the spinal cord. Furthermore, MSCs from BMA can enhance the expression of neurotrophic factors and promote the activation of endogenous neural progenitor cells, potentially fostering remyelination and neuronal repair [69]. These interactions suggest a direct mechanism by which BMA could interrupt the pathological progression of SCI and support recovery, aligning with the observed decrease in neuropathic pain and the improvement in motor functions documented [33,70,71]. This integrative role of BMA in modulating both the immune response and supporting neuroregeneration offers a dual therapeutic potential, aligning with the current understanding of SCI pathophysiology and providing a substantive hypothesis for the ongoing clinical investigations into its efficacy. These mechanisms warrant further exploration to define optimal administration strategies and long-term outcomes of BMA therapy in SCI patients.

## 3. Conventional Treatment Approach

Recently, rehabilitation therapies for spinal cord trauma have significantly evolved with the emergence of new techniques, including physical therapies that allow patients to live independent lives and the possibility of social reintegration [72]. Within the spectrum of therapies that have been implemented for the management of SCI, the use of orthobiologics and cell therapies have emerged as effective and viable alternatives for several diseases beyond the musculoskeletal system [73]. Studies using cell replacement strategies in scaffolds have shown extremely promising results; however, functional recovery still lacks reliable results and reproducibility [74]. Therefore, the possibility of a “cure” (complete regeneration and functional recovery) remains uncertain. Individuals with incomplete SCI (without complete motor paralysis) who are able to partially contract some muscle groups may benefit from neurorehabilitation protocols involving activity-based therapies to promote neural network reorganization with modest functional gains [75]. The use of robotic orthoses to assist the mobility of these patients is another alternative. However, it is expensive and therefore less available to the general population [76]. New studies have shown that epidural stimulation may represent a valid therapy to achieve functional improvement in these patients, even in cases of previous complete paralysis [77].

On the other hand, continuous research has contributed to the emergence of new techniques and treatment modalities, such as autologous orthobiologic therapy. Among the popular alternatives, BMA stands out for its significant regenerative potential, especially in complex orthopedic disorders. This biological material contains a wide variety of cells and molecular agents that play vital roles in effectively modulating inflammation and contributing to progression to the resolution phase of wound repair [17]. More detailed information regarding BMA is presented in the section immediately below.

While recent advances in rehabilitation therapies have improved the quality of life for individuals with SCI, significant limitations persist, particularly in the realm of biological recovery and neuroregeneration. Current interventions such as physical therapy, robotic orthoses, and epidural stimulation predominantly focus on compensating for lost functions rather than restoring the underlying neural structures. These treatments often fail to halt or reverse the neurodegenerative processes following SCI, leaving a substantial gap in treatment outcomes, especially for patients with complete paralysis. Additionally, the high cost and limited accessibility of advanced mechanical aids underscore the need for more broadly accessible and biologically focused therapies. BMA addresses these critical gaps by leveraging its regenerative potential to not only modulate inflammatory responses but also promote the regeneration of neural tissues and support functional recovery at a molecular level [16,78,79,80]. Unlike traditional therapies, which are mostly palliative and adaptive, BMA offers a therapeutic strategy aimed at the root causes of neurodegeneration observed in SCI, providing a foundation for more definitive and sustainable recovery outcomes.

## 4. Bone Marrow Aspirate

BMA is a well-known orthobiologic product that carries a wide variety of cells, small tissue fragments, and peripheral blood [81,82]. It is renowned for the presence of MSCs, which were first discovered in 1976 by Friedenstein [83]. These cells play an important role in tissue repair and modulation via the secretion of cytokines and growth factors with paracrine and autocrine effects [16]. MSCs in the bone marrow have the potential to differentiate into various tissues of mesodermal origin [84]. In addition to MSCs, there are also HSCs, endothelial progenitor cells, and other precursor cells, and many molecular agents, including bone morphogenetic proteins (BMPs), platelet-derived growth factor (PDGF), TGF-β, VEGF, interleukin-8 (IL-8), and IL-1 receptor antagonist (IL-1Ra), among other molecules, help tissue regeneration [16].

Due to its versatility, BMA allows collection and application to be performed in a single procedure without the need for laboratory processing, thus reducing costs and regulatory compliance issues [16]. One of the main BMA collection sites is the posterior iliac crest, as it offers a greater number of viable cells [85]. However, the quality of the aspirate is dependent on the technique used for collection: while the first 4–5 mL of BMA contains high-quality MSCs, withdrawing larger volumes leads to the dilution of the aspirate with peripheral blood [86]. The collection of small volumes in various subcortical areas appears to be more effective, resulting in a greater yield of stem cell progenitors [87]. Physicians may choose fluoroscopy or ultrasound guidance during the aspiration procedure. Anatomic landmarks are suitable for the average patient, while ultrasound guidance is recommended for larger individuals. For an average patient, an ideal volume ranging from 90 to 120 mL is suggested, while volumes exceeding 120–150 mL should be avoided in larger individuals [88]. In terms of contraindications, alternative options should be explored for patients with anemia, systemic infection, active hematologic neoplasms, or who cannot be positioned anatomically for the procedure. It is crucial for patients to disclose any medications they are taking before the procedure. Examples include prednisone, known for its anti-anabolic effects; statins, which negatively impact cell proliferation; and nonsteroidal anti-inflammatory drugs, as they interfere with platelet aggregation and clot function [89].

Among the variations in bone marrow-derived products, the BMA clot is a potential candidate for tissue regeneration. Although interpreted as a technical complication by some, it is worth noting that the coagulation property of BMA may play a relevant role in musculoskeletal tissue regeneration [90]. Clot formation occurs due to platelet activation and degranulation, which leads to the release of osteotropic cytokines and growth factors, including PDGF, EGF (epidermal growth factor), FGF (fibroblast growth factor), and TGF-β [90]. Furthermore, the BMA clot can also increase the stability of the cell graft to the injured site [90]. The fibrinolytic system, which is intrinsically tied to clot formation and degradation, also plays a vital role in the healing cascade since it leads to the secretion of angiogenic factors, an essential process for the initiation of tissue repair [91].

These biological agents synergistically contribute to the regulation of the extracellular environment where BMA is applied, modulating the inflammatory response of the injured region and inducing the proliferative response to establish tissue repair. Researchers have also explored the role of IL-1Ra, a major cytokine found in the bone marrow. This competitive antagonist binds to IL-1B and IL-1a cell surface receptors, inhibiting catabolic reactions and inflammatory effects induced by IL-1 [92]. IL-1Ra may reduce matrix degradation since IL-1B is known to upregulate MMP-3 and TNF-a gene expression, prostaglandin E2 secretion, chondrocyte apoptosis, and the suppression of collagen deposition [93].

For these reasons, BMA has been highly appreciated and investigated in several medical areas with the intention of promoting tissue regeneration of bones, cartilage, and soft tissues [94]. Although the number of studies is limited, they seem to indicate that BMA administration is a promising technique supporting the regenerative potential of the bone marrow.

The technique for collecting BMA is critical to maximizing the concentration and viability of the aspirate, which are key factors in its effectiveness for regenerative therapies. To optimize the quality of BMA, a precise technique must be followed, starting with the choice of aspiration site and the needle gauge used. The posterior iliac crest is preferred due to its accessibility and high concentration of bone marrow cells. A smaller gauge needle is typically recommended to reduce the shear stress on cells during aspiration, which can significantly affect cell viability [95]. It is also crucial to perform the aspiration slowly and steadily to minimize the dilution of bone marrow with peripheral blood, which can occur if the aspiration is too rapid or if excessive pressure is applied. Furthermore, the volume of BMA extracted in a single draw should be limited to ensure high cell density. The first few milliliters of the aspirate are typically the richest in stem cells, and subsequent draws may yield progressively diluted samples. Therefore, multiple small-volume aspirations from different sites within the iliac crest are preferred over a single large-volume draw. This technique not only increases the yield of progenitor cells but also maintains their functional viability. For the preparation and administration of BMA, minimal manipulation is key to preserving the biological characteristics of the aspirate. The BMA should be handled gently to avoid cellular damage and should not be exposed to temperature extremes or unnecessary centrifugation, which can alter cell integrity [95]. Before administration, it is critical to ensure that BMA is evenly mixed to prevent the settling of cells, which can lead to inconsistent dosages upon injection. The administration itself should be carried out using real-time guidance, such as ultrasound or fluoroscopy, to ensure the precise placement of BMA into the target tissue, optimizing the regenerative potential at the site of injury. In cases where higher concentrations of specific cell types are required, such as for more severe injuries or degenerative conditions, advanced techniques like centrifugation to concentrate bone marrow cells or the addition of biocompatible scaffolds to support cell attachment and growth may be employed. These methods can enhance the therapeutic potential of BMA by increasing the local concentration of regenerative cells and factors at the injury site, thus potentially improving clinical outcomes.

## 5. Inflammatory Modulation

“Inflammomodulation” is perhaps one of the greatest attributes of MSCs [96]. In SCIs, microglial activation promotes the release of several proinflammatory cytokines, such as TGF-β, basic fibroblast growth factor, IL-6, and IL-1 [97]. A reduction in the inflammatory response after SCI mediated by MSCs occurs mainly through the release of exosomes [98]. Exosomes are vesicular membrane structures approximately 40–120 nm in diameter produced by MSCs that act via the paracrine pathway in intercellular communication [98]. These extracellular vesicles (EVs) are involved not only in cell proliferation but also in the inhibition of apoptosis and inflammation, which are essential for complete tissue repair [98]. In the context of SCI, the microRNA present in exosomes from bone marrow-derived MSCs (BMSCs) can induce macrophage M2 polarization through the NF-κB pathway [99]. It also downregulates the expression of inducible nitric oxide synthase (iNOS), IL-6, and TNF-α, which are notable downstream mediators of inflammation, especially in the musculoskeletal system [99]. Furthermore, neuronal differentiation and axon regeneration are also elicited by BMSCs via either the inhibition or activation of specific signaling pathways [99].

## 6. Axonal Regeneration

As previously described, glial scarring hinders axonal regeneration and impairs nervous system transmission and recovery [100]. Astrocytes are the most abundant glial cells in the central nervous system and carry great responsibility because they regulate blood flow and maintain neuronal integrity and homeostasis [101]. In an eventual SCI, activated microglia convey a more neurotoxic phenotype to astrocytes [99]. The glial scars formed due to traumatic injury also prevent regenerated axons from entering the colloidal scar structure, thus making it difficult to establish neural networks [102]. However, MSC-derived exosomes can impede neurotoxic astrocyte activation, neural apoptosis, and subsequent glial scar formation [102]. By manipulating astrocyte morphology, astrocytes provide a more favorable microenvironment for axonal regeneration and functional recovery, at least in mice [103]. Moreover, MSCs can also inhibit glial scarring through diminished inflammatory cytokine levels, especially TGF-β. This cytokine is a known mediator of glial scarring since it activates the Smad signaling pathway in astrocytes [104]; however, BMSCs can reduce glial scarring and promote axon regeneration by blocking the TGF-β/Smad interaction [105].

Brain-derived neurotrophic factor (BDNF) and nerve growth factor β (β-NGF) are two of the many proteins secreted by MSCs that help to sustain neuronal survival and axonal regeneration [106]. In an animal model of SCI, MSCs reportedly expressed neural biomarkers such as βIII tubulin, enolase-2, and microtubule-associated protein 1b (MAP1b) [107]. βIII tubulin, in particular, has been acknowledged as a key component of neuronal microtubules and is responsible for axon orientation, maturation, and maintenance [99]. In a study of the transplantation of modified MSCs into SCI mice, Zhou and colleagues reported the recovery of neurological function. The authors postulated that MSC-EVs can effectively communicate with neuronal cells and activate the MEK/ERK pathway, increasing tubulin expression [108].

BMSCs can also secrete IGF-1, a growth factor that sustains their own survival. IGF-1 suppresses oxidative stress at the site of SCI, reducing the accumulation of noxious substances such as reactive oxygen species (ROS) [99]. IGF-1 reportedly targets ROS around the injury site, blocking cellular oxidative stress by inhibiting the Foxo3 transcription factor [109]. In the context of SCI, IGF-1 plays an important role in nerve cell regeneration: it binds to its corresponding receptors on neural cells, promoting neural stem cell recruitment and differentiation [110].

The PI3K/AKT/mTOR signaling pathway is indispensable for mammalian growth [111]. In vitro experiments have shown that the blockade of this pathway aggravates neuronal damage [112]; however, the transplantation of BMSCs, in particular, can restore motor function following SCI via PI3K/AKT/mTOR pathway activation [113]. An in vitro study with BMSC-EV-loaded hydrogels demonstrated that these vesicles are able to activate the PI3K/AKT/mTOR pathway and promote neuronal differentiation and axonal regeneration [112].

Another vital component is the LIN-12/NOTCH signaling pathway, which controls neural stem cell proliferation and differentiation [114]. In the event of neural damage, LIN-12 can limit the regeneration of nerve axons through autophagy, a cellular process that degrades damaged cytoplasmic materials and organelles [115]. By inhibiting the NOTCH signaling pathway, MSCs promote neuron differentiation. Additionally, the MSC-mediated activation of the Wnt/β-catenin signaling pathway plays a key role in functional recovery and axon regeneration. The Wnt family of glycoproteins is involved in neurodevelopment, axon guidance, cell proliferation, and nerve cell survival [116]. In parallel with these findings, Yoon et al. reported that the transplantation of these cells after SCI enhances axon regeneration by increasing WNT3A protein expression [117].

## 7. Angiogenesis

Traumatic SCI is known to cause the direct and significant destruction of blood vessels around the spinal cord, affecting BSCB integrity. The destruction of blood vessels in the BSCB results in ischemic necrosis, which is aggravated by secondary injury [118]. In fact, it is important to treat the vascular component in SCI patients since it contributes to the restoration of motor functions [119]. The migration of MSCs into the injured area also induces the endogenous differentiation of cells into new pericytes, thus sustaining angiogenesis [120]. A recent study confirmed that vascular endothelial cells in SCI can collect MSC-EVs, whereas in their normal state, they cannot [121]. The exact underlying mechanisms remain unclear; however, it is speculated that necrotic vascular endothelial cells may be found in a specific state that favors the reception of MSC-EVs [121]. Furthermore, the recovery of the vascular endothelium has been shown to inhibit the formation of fibrotic scars, as endothelial cells engulf myelin debris [122].

Several MMPs are released after SCI and subsequently attack the BSCB via ECM degradation [123]. These proteolytic enzymes cause BSCB destruction and the imminent penetration of inflammatory factors and neurotoxic substances, which escalates damage [124]. Researchers confirmed that MSCs maintain BSCB integrity by blocking the production of MMPs and upregulating tight junction and adherens junction protein expression [125].

## 8. Biological Neuromodulation of the Sacral Hiatus

The sacral hiatus, originating from the incomplete fusion of the lamina and spinous process of the last sacral vertebra, marks the end of the sacral canal. This hiatus is laterally delimited by two sacral horns and can usually be identified by a small depression between them. In the posterior region, the sacral hiatus area is protected by skin tissue, layers of subcutaneous fat, and the sacrococcygeal ligament. The sacral hiatus provides access to the caudal epidural space [126]. The sacral hiatus contains inferior sacral and coccygeal nerve roots. The coccygeal plexus is formed as an anastomosis between S4, S5, and the coccygeal nerve. Depending on age, the termination of the thecal sac varies between the lower edge of the S1 foramen in adults and the S3 foramen in children. In 1–5% of patients, the dural sac ends at S3 or below that level, an important fact to remember when placing the epidural needle to avoid dural puncture [126]. Variations in the anatomy of the sacrococcygeal area (up to 10%) or even a total absence of the posterior wall of the sacral canal can make it difficult to identify the anatomy of this region [127].

Ultrasound-guided BMA injection into the sacral hiatus may improve safety. Although fluoroscopy is traditionally considered the “gold standard” technique for performing caudal epidural injections, the use of ultrasound has proven to be extremely efficient in accurately guiding the needle into the caudal epidural space, achieving treatment results that may rival those obtained with fluoroscopy [128].

At our clinic, ultrasound examinations are performed with the patient lying in the prone position. Vigorous asepsis is performed, and gauze is placed in the intergluteal region for comfort and protection. The area is examined with the aid of ultrasound with a 7–16 MHz multifrequency linear transducer in lean patients and in obese patients, and a low-frequency multifrequency convex transducer is used to identify the anatomy of the sacral hiatus and its possible anatomical variations. The transducer is first placed transversely in the midline to obtain a transverse ultrasound view of the sacral hiatus, and then the transducer is rotated 90°, placing the image on the longitudinal axis. In the transverse axis, the two inverted U-shaped hyperechoic structures are the two bony prominences of the sacral horns. Between the two horns, there are two band-shaped hypoechoic structures that form the sacrococcygeal ligament. The linear hyperechoic structure at the bottom is the dorsal bony surface of the sacrum. The sacral hiatus is the hypoechoic region observed between the two sacral horns and below the sacrococcygeal ligament [126]. After identifying the anatomical landmarks, an anesthetic block of the skin and sacrococcygeal ligament is performed with 1% lidocaine, which can be performed with a 22 g spinal needle in obese patients or a simple 25 × 8 mm needle in lean patients. When passing through the sacrococcygeal ligament, a loss of resistance is felt. The appropriate positioning of the needle can be confirmed by injecting a small amount of the solution under Doppler ultrasound vision [129]. Once inside the sacral hiatus, the needle appears as a hyperechoic structure on ultrasound. The Doppler effect manifests inside the canal and confirms correct positioning. It is also possible to observe the movement of the sacrococcygian ligament in real time during infiltration, even in B mode. Throughout the whole procedure, the patient is assisted by physicians and nurses to ascertain safety and comfort. After the infiltration was complete, the area was cleaned, and Blood Stop^®^ (AMP, São Paulo, Brazil) was applied.

The intricate network of neural communication between the sacral plexus and the lumbar plexus supports our hypotheses. The stimulation of both plexuses is associated with communication with the chain of sympathetic ganglia present in the spine in its bilateral anterolateral portion.

It connects this entire complex system with the vagus nerve complex and from there to the higher centers, since up to 80% of the vagal fibers are afferent and mostly unmyelinated [130]. The parasympathetic premotor neurons that project to preganglionic neurons originate in a medullary nucleus called the dorsal motor nucleus of the vagus nerve, where the effector response arises in the ganglion close to the target organ and then moves to the target organ [131]. The vagus nerve contains numerous general sensory afferent fibers that regulate basic sensations, including touch, pain, temperature, and pressure, and more sophisticated sensory fibers, such as visceral sensory afferent fibers, somatic afferent fibers, and visceral efferent fibers [132].

We believe that the vagal system receives an important stimulus from bioneuromodulation since vagal afferents ascend with information to an important visceral sensory nucleus located in the medulla called the nucleus of the solitary tract (NTS) [133]. NTS is responsible for an important integrative function of signals coming from the viscera. From this nucleus, a series of interneurons project to important nuclei in the spinal cord, brainstem, hypothalamus, and forebrain. They also receive synaptic inputs from all these areas [134]. Taking this into consideration, it is possible that vagal efferent fibers are also stimulated. Given that descending impulses are present in both the locomotor system and sympathetic system and that sympathetic output is increased in response to locomotion [135], it is likely that there is cellular connectivity mediating the two systems (Figure 2).

Integration between locomotor and sympathetic centers is observed in the brainstem, but this communication is lost after injury [136]. However, there have been cases where the ascending control of the lumbar spinal segments can coordinate and integrate movement without the brainstem spinal centers [38]. After SCI, both sympathetic and parasympathetic functions are significantly impaired due to the loss of communication between the autonomic centers and the spinal sympathetic flow [38]. Therefore, it is possible that biological stimuli from BMA at the spinal cord ascend via the sacral hiatus. The stimulation of the sacral and lumbar plexus centers may occur via the anterolateral prevertebral sympathetic ganglionic chain, which ascends through the vagus nerve to the superior centers. Sympathetic preganglionic neurons located in the lamina intermediaria of the T1-L2 spinal cord retain spontaneous activity, and neuromodulation can increase thoracic sympathetic output, improving cardiovascular function after cervical SCI [137]. This finding reinforces our hypothesis to explain the expected benefits of bioneuromodulation on the bladder and bowel functions of patients subjected to this method.

## 9. Conclusions

In patients with SCI, the objective of any therapy would be the restoration of partial or total movement to reduce psychosocial burden and additional complications that may arise from a lack of exercise or limited movement. The success of SCI treatment strategies is heavily influenced by cell activity and communication within the niche after injury, especially between resident stem cells, whose activation, recruitment, and modulation may contribute to recovery.

Here, we propose that the biological neuromodulation of the sacral hiatus with BMA might be achieved via the activation of preganglionic neurons in the lamina intermediaria of the spinal cord (T1-L2). This could promote beneficial effects on the functions innervated by postganglionic neurons, and their sympathetic stimulation correlated with bowel and bladder functions, for example.

Regenerative cellular therapies, including BMA injections, may represent a promising alternative for complex tissue injuries, as these cells can survive after transplantation and migrate to injured areas. It is possible that the wide array of biochemical cues provided by BMA at the sacral hiatus ascends and stimulates the lumbar centers, propagating signals that activate higher centers. Considering these factors, BMA may reverse the disruption of neurological function and communication following SCI.

## Figures and Tables

**Figure 1 bioengineering-11-00461-f001:**
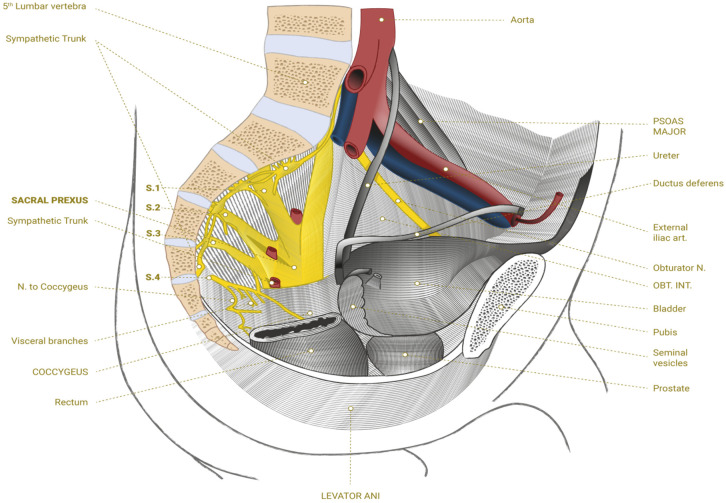
Sacral plexus.

**Figure 2 bioengineering-11-00461-f002:**
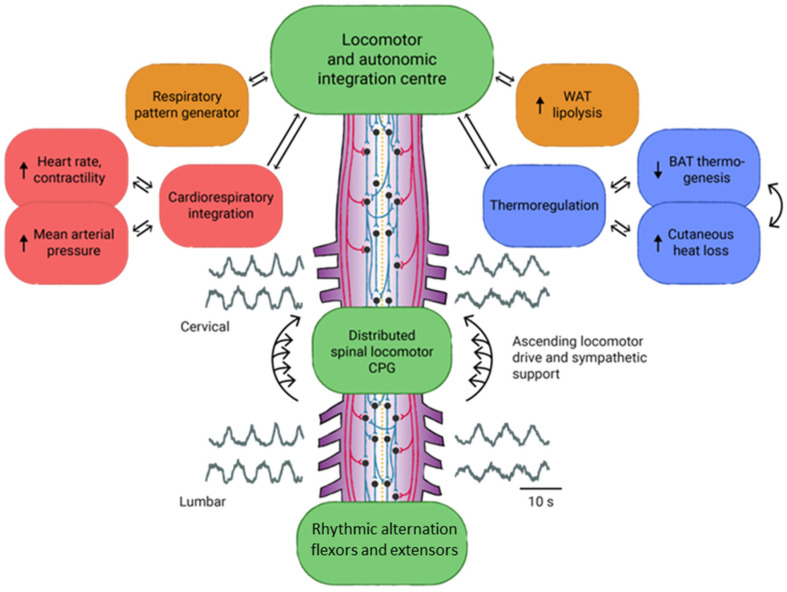
Medullary spinal control center for the integration of sympathetic autonomic support.

**Table 1 bioengineering-11-00461-t001:** Typical growth factors found in BMA.

Growth Factor	Biological Role
Platelet-Derived Growth Factor (PDGF)	Promotes cell proliferation, angiogenesis, and extracellular matrix formation; Stimulates the migration and proliferation of numerous cells involved in wound healing, including fibroblasts, endothelial cells, and smooth muscle cells [23].
Transforming Growth Factor-Beta (TGF-β)	Regulates cell growth, differentiation, and matrix production; Promotes the synthesis of collagen and other components of the extracellular matrix, thus contributing to tissue remodeling and wound closure; Regulates the immune response and conveys anti-inflammatory properties [24].
Vascular Endothelial Growth Factor (VEGF)	Potent inducer of angiogenesis as well as the proliferation and migration of endothelial cells, leading to the formation of new capillaries in the wound bed [25].
Fibroblast Growth Factor (FGF)	Promotes fibroblast proliferation and migration, which are essential for collagen synthesis and tissue remodeling; Triggers angiogenesis, epithelial cell migration, and the production of extracellular matrix components [26].
Insulin-like Growth Factors (IGFs)	Regulate cell proliferation, differentiation, and tissue repair; Promote the synthesis of collagen and other matrix proteins, dictate cell survival, and elicit anti-apoptotic effects [27].
Stem cell factor (SCF)	Regulates hematopoietic stem cells within the stem cell niche in the bone marrow; Enhances the survival of HSCs in vitro and plays a crucial role in the self-renewal and maintenance of HSCs in vivo [28].
Stromal cell-derived factor-1 (SDF-1)	Guides the migration of stem and progenitor cells, including endothelial progenitor cells [29].

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
