# Peer review of "Sacral Bioneuromodulation: The Role of Bone Marrow Aspirate in Spinal Cord Injuries"

_bioengineering, 2024, doi:10.3390/bioengineering11050461_

Round 1

Reviewer 1 Report

Comments and Suggestions for Authors

The article titled "Sacral Bioneuromodulation: The Role of Bone Marrow Aspirate in Spinal Cord Injuries" presents an intriguing hypothesis on the potential benefits of bone marrow aspirate (BMA) for treating spinal cord injuries (SCIs). The hypothesis suggests that BMA, rich in various cell populations and bioactive molecules, could modulate inflammation and pain while promoting neural repair through its application at the sacral hiatus. This innovative approach aims to stimulate lumbar centers and propagate signals to higher structures, potentially reversing neurological dysfunction caused by SCIs.

Strengths:

·       Comprehensive Literature Review: The article thoroughly reviews the current understanding of SCIs, including the pathophysiology, conventional treatments, and limitations of existing therapies. This foundation effectively highlights the necessity for novel treatments.

·       Innovative Hypothesis: The proposition that BMA can act as a neuromodulatory agent via the sacral hiatus is novel and intriguing. It opens a new avenue for research into regenerative therapies for SCIs.

·       Detailed Mechanistic Insights: The authors provide an extensive overview of how BMA could influence various biological pathways, including inflammation modulation, axonal regeneration, and angiogenesis. These discussions are supported by existing research findings, lending credibility to the hypothesis.

But for enhancing the publishability of the article "Sacral Bioneuromodulation: The Role of Bone Marrow Aspirate in Spinal Cord Injuries" and its acceptance in a journal, several improvements are recommended. These include deepening certain sections, clarifying methodology, and adding information about previous research. Here are specific suggestions:

·       Expanding Theoretical Foundation

Pages 2-3, "Introduction" section: Include more comprehensive information about previous studies on the use of BMA for treating SCI. How does your review build upon or diverge from these findings? Highlighting unique contributions or differentiating factors of your work could strengthen the rationale for the review.

·       Methodological Clarifications

Page 6, "Pathophysiology of Spinal Cord Injuries" section: Although the pathophysiology is well-described, the connection between this understanding and the hypothesis that BMA could influence these pathways needs further elaboration. Specific mechanisms through which BMA is hypothesized to affect the recovery process could be more clearly outlined.

Page 7, under "Conventional Treatment Approach": Detailing any specific limitations or gaps in current treatment approaches that BMA directly addresses would help to clarify the significance of the study's approach.

Throughout the document, there are opportunities to refine the writing for clarity and conciseness. Ensuring technical terms are well-defined at first use and maintaining a consistent narrative flow would improve readability.

·       Citation and Reference Expansion

Throughout the document: Increase the number of citations to relevant literature, especially in sections discussing the background and rationale for using BMA in SCI treatment. This not only supports the argument but also situates the research within the current scientific discourse. Some key publish are missed, such as:

doi: 10.1007/s10571-006-9007-2

doi: 10.3390/biom9120811 

·       Technical Details

Page 8, "Bone Marrow Aspirate" section: Offer more detailed descriptions of the BMA collection, preparation, and administration procedures. This could include addressing any variations in technique that might affect the concentration or viability of the aspirate.

Incorporating these improvements could significantly strengthen the article's contribution to the field and its appeal to a journal's editorial board and reviewers.

Author Response

All the comments were duly addressed in the manuscript for your kind perusal

Reviewer 2 Report

Comments and Suggestions for Authors

Authors should explain the topic of the review in detail in the abstract section

Comments on the Quality of English Language

The English language should be improved

Author Response

(The authors gave the same response as above.)
